

# Biological response of eelgrass epifauna, Taylor's Sea hare (*Phyllaplysia taylori*) and eelgrass isopod (*Idotea resecata*), to elevated ocean alkalinity

Kristin Jones[1], Lenaïg G. Hemery[1], Nicholas D. Ward[1-2], Peter J. Regier[1], Mallory Ringham[3], Matthew D. Eisaman[4-5]

[1] Coastal Sciences Division, Pacific Northwest National Laboratory,1529 W Sequim Bay Rd, Sequim, WA, USA
[2] School of Oceanography, University of Washington, Seattle, WA, USA
[3] Ebb Carbon, Inc., San Carlos, CA, USA
[4] Department of Earth & Planetary Sciences, Yale University, New Haven, CT, USA
[5] Yale Center for Natural Carbon Capture, Yale University, New Haven, CT, USA

*Correspondence to*: Kristin Jones (Kristin.Jones@pnnl.gov), Lenaig Hemery (Lenaïg.Hemery@pnnl.gov)

**Abstract.** Marine carbon dioxide removal (mCDR) approaches are under development to mitigate the effects of climate change with potential co-benefits of local reduction of ocean acidification impacts. One such method is ocean alkalinity enhancement (OAE). A specific OAE method that avoids issues of solid dissolution kinetics and the release of impurities into the ocean is the generation of aqueous alkalinity via electrochemistry to enhance the alkalinity of the surrounding water and extract acid from seawater. While electrochemical acid extraction is a promising method for increasing the carbon dioxide sequestration potential of the ocean, the biological effects of this method are relatively unknown. This study aims to address this knowledge gap by testing the effects of increased pH and alkalinity, delivered in the form of aqueous base, on two ecologically important eelgrass epifauna in the U.S. Pacific Northwest, Taylor's sea hare (*Phyllaplysia taylori*) and eelgrass isopod (*Idotea resecata*), across pH treatments ranging from 7.8 to 9.3. Four-day experiments were conducted in closed bottles to allow measurements of the evolution of carbonate species throughout the experiment with water refreshed twice daily to maintain elevated pH. Sea hares experienced mortality in all pH treatments, ranging from 40% mortality at pH 7.8 to 100% mortality at pH 9.3. Isopods experienced lower mortality rates in all treatment groups, which did not significantly increase with higher pH treatments. Different invertebrate species will likely have different responses to increased pH and alkalinity, depending on their physiological vulnerabilities. Investigation of the potential vulnerabilities of local marine species will help inform the decision-making process regarding mCDR planning and permitting.

## 1 Introduction

Among the many other impacts of climate change, increased levels of atmospheric carbon dioxide ($CO_2$) drive global decreases in ocean pH and calcium carbonate ($CaCO_3$) saturation states (Doney et al., 2009; Doney et al., 2020) known as ocean acidification (OA), posing a significant threat to marine organisms and ecosystems (Doney et al., 2020). These increased $CO_2$



levels are primarily caused by human sources, such as fossil fuel combustion, cement production, and land use changes (Doney, et al., 2009; Orr et al., 2005). OA can lead to various detrimental effects on marine life, including decreases in survival, growth, calcification, development, and abundance, particularly for slow moving or sessile animals (Kroeker et al., 2013). Marine

invertebrates can experience physiological effects such as oxidative stress, decreased immunity, decreased growth and development, and lower reproductive success (Shi and Li, 2023). OA can be particularly harmful to organisms in early life stages, affecting fertilization, larval development, dispersal, and settlement (Ross et al., 2011). Moreover, OA can negatively impact food web dynamics and ecosystem processes (Fabry et al., 2008).

Natural carbon sinks on land and in the ocean help to reduce atmospheric $CO_2$ but are not keeping pace with increasing

anthropogenic emissions, prompting research efforts to explore methods of enhancing the ocean's natural carbon sink through marine carbon dioxide removal (mCDR) (National Academies of Sciences, Engineering, and Medicine, 2022). One method of mCDR is an ocean alkalinity enhancement (OAE) approach that electrochemically processes salt (e.g., sodium chloride NaCl) to generate aqueous acid (hydrochloric acid HCl), which is removed from the system, and base (sodium hydroxide NaOH), which is mixed with the seawater stream and returned to the ocean, thus enhancing the alkalinity of the surrounding water (de

Lannoy et al., 2018; Eisaman et al., 2018; Eisaman et al., 2023; Lu et al., 2022; Ringham et al., 2024; Tyka et al., 2022; Wang et al., 2023). The increased surface alkalinity drives additional ocean uptake of atmospheric $CO_2$, which is ultimately stored in seawater as dissolved bicarbonate (Cross et al., 2023, Eisaman et al., 2023, Ringham et al., 2024). While OAE could be a promising avenue for reducing atmospheric $CO_2$, the biological response of marine organisms and impact on ecosystems of locally increasing pH and alkalinity remains largely unknown.

Changes in ocean pH can have implications for marine life and the health of marine ecosystems. pH shifts can affect physiology of aquatic organisms by disrupting acid-base regulation essential for cellular function, can inhibit fixation and respiration of $CO_2$, and reduce nutrient uptake (Tresguerres et al. 2020, Tresquerres et al. 2023). Multicellular marine organisms rely on intracellular and extracellular pH gradients and modulation for metabolic processes. This is regulated through an acid-base balance, which can be disrupted if environmental conditions, such as pH or $CO_2$, are altered (National Research Council,

2010). Many organisms have the ability to control their internal pH to an extent, but some may be able to acclimate better than others at the cost of high metabolic demand (Portner et al. 2000). Previous studies have explored the impact of pH changes, particularly in the context of OA, on a variety of marine organisms, but only a few have studied the impacts of increasing pH. A method used in aquaculture to reduce biofouling involves increasing local alkalinity with the addition of calcium hydroxide (Comeau et al. 2017). In this study, bivalves were quickly exposed to a 12.7 pH solution and exhibited short-term behavioral

stress, but did not show any mortality, likely due to the quick dispersal of the alkaline solution. The bivalves were then exposed to weakly elevated pH (9.2) consistently for three days, in which they experienced prolonged closure of their valves, indicating an "avoidance behavior," however the behavior ceased when treatment was completed, and no mortality was observed (Comeau et al. 2017). Another study investigated the effects of increased ocean alkalinity on red calcifying algae. The algae experienced a 60% increase in carbonate production when alkalinity was increased from 2694 µEq $L^{-1}$ to 3454 µEq $L^{-1}$

(resulting in a pH increase from 7.97 to 8.2), but these alkalinity and pH increases had no significant negative impacts on



primary productivity, respiration or photophysiology (Gore et al., 2019). In a controlled laboratory experiment, European green crabs were exposed to calcium hydroxide to determine biological effects of increased alkalinity. Calcium hydroxide was added in two concentrations (0.28 mmol $L^{-1}$ and 0.54 mmol $L^{-1}$), to raise the $pH_{NBS}$ (8.0 - 8.7). The green crabs experienced physiological disruption in acid-base regulation, respiratory alkalosis and hyperkalemia (Cripps et al., 2013). Few studies have

investigated biological response to enhanced ocean alkalinity, but this research is needed to move this field forward, following best practices for collaborative OAE research (Oschilies et al. 2023).

In the Pacific Northwest region of the U.S. and Canada, eelgrass is critical to many nearshore ecosystems, playing key roles such as providing habitat for other species and acting as a food source, directly and indirectly to support food webs (Thayer & Phillips, 1977). In addition, eelgrass ecosystems provide a variety of supporting services, such as refuge, nursery habitat,

foraging areas, and habitat areas for reproduction, as well as regulating services, such as shoreline protection, sediment stability, water quality improvement, and climate change regulation (Sherman & DeBruyckere, 2018). Pacific salmon, both a culturally and commercially important species, rely on valuable eelgrass habitat. This ecosystem provides foraging opportunities for juvenile salmon that promote growth and survival during their critical early life stage (Kennedy et al., 2018). However, Pacific Northwest eelgrass ecosystems are at risk from a variety of threats, including invasive species, anthropogenic

contaminants, and global shifts in temperature and sea level rise (Sherman & DeBruyckere, 2018). Although manipulation experiments have shown that acidic conditions may actually alleviate stress and promote eelgrass productivity (Zayas-Santiago et al., 2020; Zimmerman et al., 2017), the synergistic impacts on eelgrass epifauna under either acidic or alkaline treatments remains unknown.

Eelgrass isopods (*Idotea resecata*) typically range from Alaska, U.S. to California, U.S. and are found in eelgrass ecosystems.

They feed on eelgrass blades and kelp and play a significant role in food webs as a prey source for many fish species, including Pacific salmon (Bridges, 1973, Ricketts and Calvin, 1952; Welton and Miller, 1980). Taylor's sea hares (*Phyllaplysia taylori*) typically range from British Columbia, Canada to California, U.S. and spend their lives on the blades of eelgrass, feeding on epiphytic diatoms (Beeman, 1963). Sea hares are herbivores and use their green coloration and vertical stripes as camouflage from predation among the eelgrass blades (Bridges, 1973). In the eelgrass ecosystem, both isopods and sea hares graze on the

epiphytic algae, which block the eelgrass from the sun and limit photosynthesis. This grazing reduces the epiphyte load on the eelgrass blades, allowing for continued photosynthesis (Lewis & Boyer, 2014). Studies examining eelgrass mesograzer species sensitivity to environmental changes, such as pH, salinity, and temperature, found that shifts in environmental conditions are likely to affect their feeding on epiphytes, and this can lead to indirect effects on the growth and productivity of the eelgrass ecosystem (Hughes et al., 2017; Tanner et al., 2019).

Marine epifauna local to the Pacific Northwest experience substantial natural variability in pH over daily to seasonal timescales. Over the course of the year, pH in Puget Sound surface layer waters can vary by more than one pH unit with even greater variability at the numerous river outlets (e.g., 6.5 to 8.5) around the region (Bianucci et al., 2018; Fassbender et al., 2018). This variability is driven primarily by tides, diel productivity patterns, river discharge, and seasonal weather variability.



The objective of this study was to determine the biological responses of eelgrass epifauna (Taylor's sea hares and eelgrass isopods) to increased pH and alkalinity levels to inform future mCDR field trials and identify knowledge gaps pertaining to laboratory and field trials in the context of OAE. We investigated both the biological and chemical responses to the addition of NaOH to seawater to determine safe bounds of operation for OAE interventions on specific species. We contextualized the carbonate speciation conditions organisms were exposed to throughout these experiments through measurements of pH and partial pressure of $CO_2$ ($pCO_2$) and investigated animal mortality and behavioral trends. We hypothesized that prolonged exposure to pH and alkalinity outside the bounds of typical coastal variability would cause widespread mortality of the studied organisms.

## 2 Methods

### 2.1 Laboratory experiment

The experiments were conducted at the Pacific Northwest National Laboratory marine laboratory in Sequim, WA, U.S. Eelgrass has been cultivated in outdoor mesocosm tanks on the dock at this facility for over 20 years, supplied with unfiltered seawater from the mouth of Sequim Bay. pH measured from the unfiltered seawater at the facility was shown to display large variability throughout the day/year (Myers et al., in prep; Ward, personal communication). Adult Taylor's sea hares and eelgrass isopods were collected from these eelgrass ecosystem tanks in July to September, 2023. Three batches of 120 sea hares were collected from late July to early August 2023 and three batches of 120 isopods were collected from late August to late September 2023, each batch undertaking a week of acclimation before being used for the experiments. Sea hares and isopods were gently collected by hand or with nets from the outdoor eelgrass tanks and transferred to three acclimation tanks in the on-site wet laboratory. Acclimation tanks were filled with about 2.5 cm of sediment collected from one of the outdoor eelgrass tanks, and had continuously flowing raw seawater from Sequim Bay to provide both flow and nutrients to the organisms. The flow rate (not measured) was fast enough to allow for the water to rapidly refresh and maintain a temperature as close to the natural environment as possible. Animals were provided daily with eelgrass blades and diatom masses from the outdoor tanks as habitat and food sources in the acclimation tanks. The animals acclimated for one week during which time they were checked daily. Any mortality was noted, and the deceased animals were removed from the tank. At the end of acclimation, 100 sea hares and 100 isopods were randomly collected from the surviving organisms to enter the experiments. After acclimation, animals were not fed for the remainder of the experiment, based on the standards for acute toxicity tests with macroinvertebrates (ASTM, 2000).

To start the experiment and for each water change, 7 L of unfiltered seawater were gently poured into an 8-L bucket and the $pH_{NBS}$ (NBS scale), salinity, temperature, and dissolved oxygen (DO) of the seawater were measured using a YSI ProDSS probe that was calibrated daily. The control group pH was the pH of Sequim Bay water collected via the facility's seawater intake, located near the seafloor at about 10 m depth, at the time of water change (generally around 7.8 +/- 0.3 due to natural variability). Temperature and salinity readings, along with water volume, were used to calculate the amount of NaOH needed



to reach the desired pH for each treatment group, using a CO2SYS Excel macro (Pierrot et al., 2021). The low treatment group had a target pH of 8.3, the medium treatment group had a target pH of 8.8, and the high treatment group had a target pH of 9.3. Commercial aqueous NaOH (0.5 M, Honeywell Chemicals 352576X1L) was used to replicate the concentration of NaOH derived from an electrodialysis method for creating acid and base from seawater (Eisaman et al. 2023; Ringham et al., 2024).

After mixing in NaOH to the seawater, the pH level was measured using the YSI probe and adjusted on a drop-by-drop basis until within 0.1 of the target pH. Preparation of pH treatment was always completed in a separate container from the organisms to avoid exposing animals to incorrect pH levels.

Once the seawater pH was adjusted to a target value, pH, salinity, temperature, and DO were recorded (Table 1). A gas sample (process described in sect. 2.2) was taken for $pCO_2$ analysis, and the water was carefully transferred to fill to the top six 1-L

glass jars without creating bubbles. For each of the sea hares and isopods experiments, five animals were randomly chosen from the acclimation tanks and placed into five of the six jars per pH treatment. Lids were placed on the jars, which were then placed randomly on a laboratory water table (Fig. 1) to account for potential variances in environmental parameters (e.g., light, air temperature). Lids were not perforated to limit air exchange and abiotic alteration of pH (i.e., atmospheric equilibration, which would decrease the intended pH treatment and increase $pCO_2$ as $CO_2$ diffused into the seawater in response to the NaOH

addition). For each pH treatment, a sixth jar was filled with controlled seawater or treated seawater, but without organisms, as a chemical control.

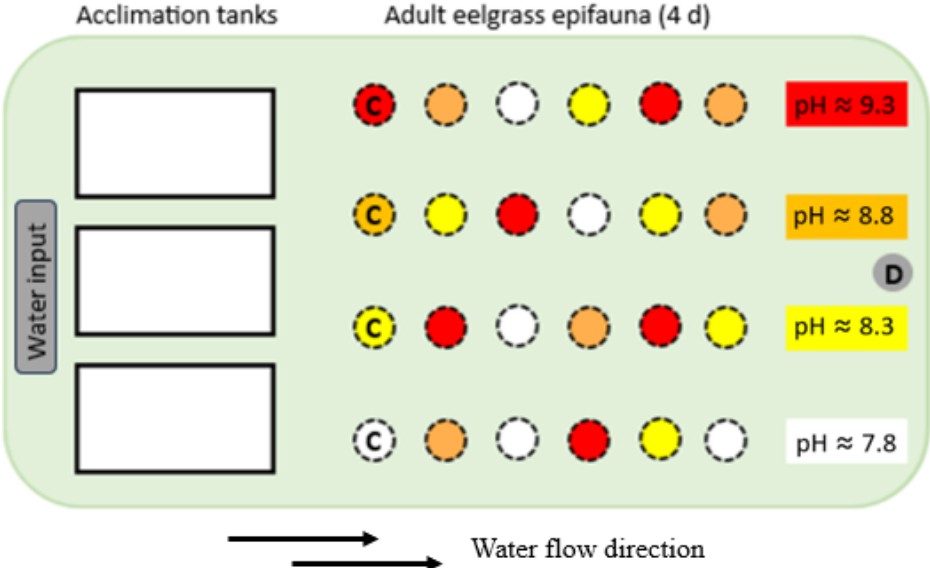

**Figure 1. Experimental design of laboratory water table, including the 1-liter glass jars (colored circles) used in the experiment and**
**the acclimation tanks (white rectangles). The white circles indicate control jars with pH ≈ 7.8, yellow circles indicate low treatment with pH ≈ 8.3, orange circles indicate medium treatment with pH ≈ 8.8, and red circles indicate high treatment with pH ≈ 9.3. The circles with "C" indicate the chemical control jars in which only treated, or control seawater was added without the presence of sea hares or isopods. The circle with "D" indicates the drain on the water table.**



This water change process was repeated twice a day to ensure proper oxygenation and pH treatment in the jars and to remove any excrement or deceased organisms. For each water change, used water from all the jars within a treatment group was carefully pooled into a single bucket, and pH, salinity, temperature, and DO within this bucket were measured and a gas sample was taken for $pCO_2$ analysis. Water quality measurements and a gas sample were also taken from the chemical control from

each treatment group. Water was refilled as described above. Organisms were checked for mortality, any casualties were removed from the jars, and any unusual behavior, such as reproduction or cannibalism among the animals, was also noted. Between water changes, a standpipe was inserted into the drain of the water table to create a water bath of about 10 cm to keep the jars at a cooler temperature akin to the natural seawater in Sequim Bay. The temperature of the water from Sequim Bay ranged from 10.9°C to 15.3°C depending on the time of day and month of year (Table 1). The experiment was conducted over

four days and was repeated three times for both sea hares and isopods, with a new batch of animals each time.

**Table 1. Range in pH, salinity, dissolved oxygen, and temperature of ambient seawater before NaOH treatment was added.**

|  | Round 1 | | | | Round 2 | | | | Round 3 | | | |
|---|---|---|---|---|---|---|---|---|---|---|---|---|
| **Sea hares** | pH | Salinity (ppt) | DO (mg/L) | Temp | pH | Salinity (ppt) | DO (mg/L) | Temp | pH | Salinity (ppt) | DO (mg/L) | Temp |
| Minimum | 7.70 | 29.87 | 7.07 | 12.90 | 7.72 | 29.89 | 7.24 | 12.40 | 7.59 | 30.55 | 6.23 | 12.00 |
| Maximum | 8.09 | 30.51 | 9.93 | 15.30 | 8.09 | 31.01 | 10.00 | 14.40 | 8.00 | 31.07 | 8.94 | 14.50 |
| Average | 7.84 | 30.16 | 8.31 | 13.74 | 7.89 | 30.28 | 8.70 | 13.45 | 7.73 | 30.90 | 6.99 | 12.56 |
| **Isopods** | pH | Salinity (ppt) | DO (mg/L) | Temp | pH | Salinity (ppt) | DO (mg/L) | Temp | pH | Salinity (ppt) | DO (mg/L) | Temp |
| Minimum | 7.65 | 31.94 | 6.27 | 12.30 | 7.62 | 32.19 | 5.80 | 12.00 | 7.59 | 32.57 | 5.52 | 10.90 |
| Maximum | 8.02 | 32.50 | 9.25 | 14.70 | 7.85 | 32.48 | 7.74 | 13.40 | 7.84 | 32.99 | 7.94 | 12.10 |
| Average | 7.81 | 32.24 | 7.22 | 13.04 | 7.71 | 32.29 | 6.56 | 12.57 | 7.67 | 32.78 | 6.34 | 11.38 |

## 2.2 pCO₂ sampling and carbonate speciation calculations

To collect gas samples from the treated seawater, a water sample was collected in a 300-mL bottle, poured gently to avoid extraneous bubbles, and poured to overflow to eliminate headspace. 60 mL of nitrogen ($N_2$) were injected with a syringe into the bottle; simultaneously, 60 mL of water were removed from the bottle with another syringe. This created a headspace of $N_2$ at the top of the bottle. After vigorously shaking the bottle for one minute to distribute the $N_2$ throughout the water sample, the gas sample was extracted using a unique plastic syringe for each sample. A gas analyzer, Picarro G2508 Cavity Ring-Down

Spectrometer with a flow limiter installed on the inlet to reduce gas flow rates, was used to measure the partial pressure of $CO_2$ present in the gas sample (e.g., Regier et al., 2023).

Calculation of carbonate speciation from measured pH and $pCO_2$ was not attempted considering these are correlated parameters and the least accurate for making such calculations. However, the goal of these measurements was not to precisely quantify



total alkalinity and DIC dynamics, but rather to provide basic context for the biological responses. Directly measuring DIC
and total alkalinity was not practical given the amount of replication, treatments, and water refreshes that were performed.

## 2.3 Data analysis

All data analyses were conducted in R Statistical Software version 4.3.1 (R Core Team, 2023). Data exploration, visualization, and analysis were completed using the following libraries: *ggplot2*, *tidyverse*, *dplyr*, *ggpubr*, *readxl*, *lmtest*, *nlme*, *rstatix* and *car* (Fox and Weisberg, 2019; Kassambara, 2022; Kassambara, 2023; Pinheiro et al. 2023; Wickham, 2016; Wickham et al.
2019; Wickham et al. 2023; Wickham and Bryan, 2023; Zeilis and Hothorn, 2002). One-way analysis of variance (ANOVA) tests were applied to mortality data from the four different treatment groups to determine whether differences between groups were statistically significant. Data from each round were pooled into one data file for statistical analysis. When p-values indicated a significant difference ($p < 0.05$), Tukey tests were applied to identify between which groups these differences appeared. Multiple linear regression models were used in conjunction with the boxplots to determine whether pH was the
driving factor behind mortality, as opposed to abnormal salinity, low DO levels, or temperature variation.

Chemical toxicity tests use a Lethal Concentration 50 ($LC_{50}$) value to measure the toxicity of a substance and its concentration that results in 50% mortality of a test subject (Government of Canada, 2024). While $LC_{50}$ tests generally measure the concentration of a particular substance, we were more interested in looking at the overall effect the pH values had on the organisms' mortality, as opposed to the actual amount of NaOH added to achieve said pH. Therefore, for each round of sea
hare and isopod experiments, the number of days at which 50% mortality occurred for each pH treatment was used as an $LC_{50}$ value.

## 3 Results

### 3.1 Change in pH and $pCO_2$

Changes in pH and $pCO_2$ were evaluated for geochemical context for the biological experiments, but were not a major focus
of this study. In the chemical control jars for the sea hare experiment, i.e., with no organisms present, the $pCO_2$ in the pH 7.8 control group (i.e., without addition of NaOH) was consistently higher both before and after the water changes than any of the other pH treatment groups (Fig. 2 top). This is consistent with the expected decrease in the $pCO_2$ resulting from the addition of NaOH to seawater. With organisms present in the pH 7.8 control group (i.e., without the addition of NaOH), there was a large increase in $pCO_2$ after the water changes. However, as pH increased, this change in $pCO_2$ appeared to decrease, even
sometimes experiencing a decrease in $pCO_2$ in the medium (pH 8.8) and high (pH 9.3) treatment groups (Fig. 2 bottom).





**Figure 2.** pCO₂ (ppm) measured in sea hare jars, before and after twice daily water changes for both jars without organisms (top), and jars with organisms (bottom). Closed symbol = start of a segment using refreshed water, open symbol = end of a segment before a water change.



Similarly to the sea hare experiment, in the isopod experiment the chemical control jars without organisms had higher $pCO_2$ levels in the control group (i.e., pH 7.8) both before and after the water changes (Fig. 3 top). With organisms present, control

group $pCO_2$ mostly increased after the water change (79% of the time), but no clear pattern was discernible within the treatment groups; no treatment group had consistently lower $pCO_2$ than the rest and all treatment groups had variable $pCO_2$ levels before and after the water changes (Fig. 3 bottom).

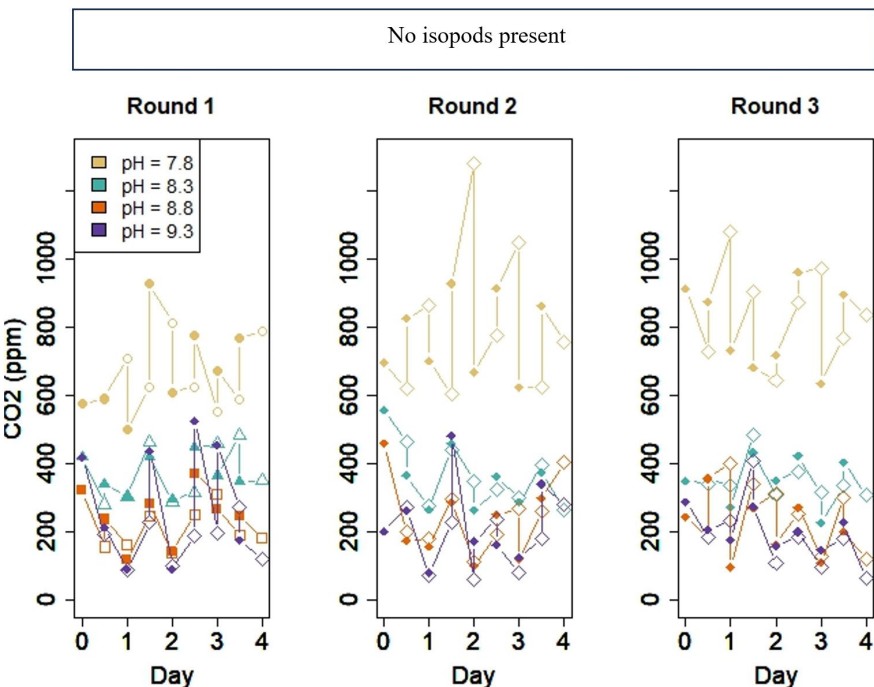






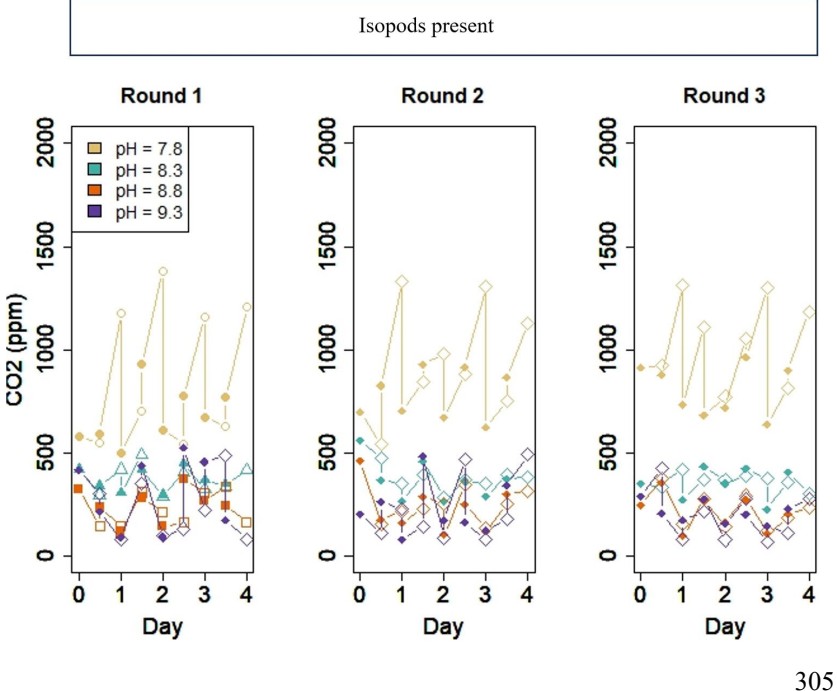


**Figure 3. pCO₂ (ppm) measured in isopod jars, before and after twice daily water changes for both jars without organisms (top), and jars with organisms (bottom). Closed symbol = start of a segment using refreshed water, open symbol = end of a segment before a water change.**


### 3.2 Animal mortality

In all three rounds of the experiment, sea hares experienced 100% mortality at pH 9.3: on the last day of experiment in the first round, after 3 days in the second round, and at 3.5 days in the third and last round. The other treatment groups never saw 100% mortality. Mortality was lowest (28%) for the control group, and mortality decreased as the pH lowered from 9.3 to 8.8, 8.3,

and 7.8 (or control group) (Fig. 4 top).

In all rounds, the ANOVA test showed significant differences between the four treatment groups (ANOVA, $p < 0.001$). Further analysis with a Tukey test (Table 2) showed that the mortality in the high treatment group was significantly larger than mortality in control, low, and medium treatment groups in the first round (TukeyHSD, $p < 0.001$, $p < 0.001$, $p = 0.02$ respectively) and third round (TukeyHSD, $p < 0.001$, $p = 0.003$, $p = 0.02$ respectively). In the second round, mortality in the

high treatment group was significantly larger than mortality in control and low treatment groups (TukeyHSD, $p < 0.001$) but not compared to the medium treatment. However, unlike the other rounds, mortality in the medium treatment group was significantly higher than that of the control treatment group (TukeyHSD, $p = 0.019$).





Isopod mortality showed similarity between treatment groups, and no clear patterns were present in any of the three rounds of experiments (Fig. 4 bottom). Mortality was observed in every treatment group in round 1 and round 2 of the experiments,

however no mortality was seen in the low treatment group in round 3. Maximum mortality never reached above 50% for any treatment group in any round of the experiment. There were no significant differences in mortality between any of the treatment groups in round 1 and 2 (ANOVA, $p = 0.104$, $p = 0.086$). In round 3, the mortality in the low treatment group was significantly less than that of the medium and high treatment group (TukeyHSD, $p < 0.001$, $p = 0.02$; Table 2). The medium treatment group also had significantly higher mortality than the control group (TukeyHSD, $p = 0.008$).


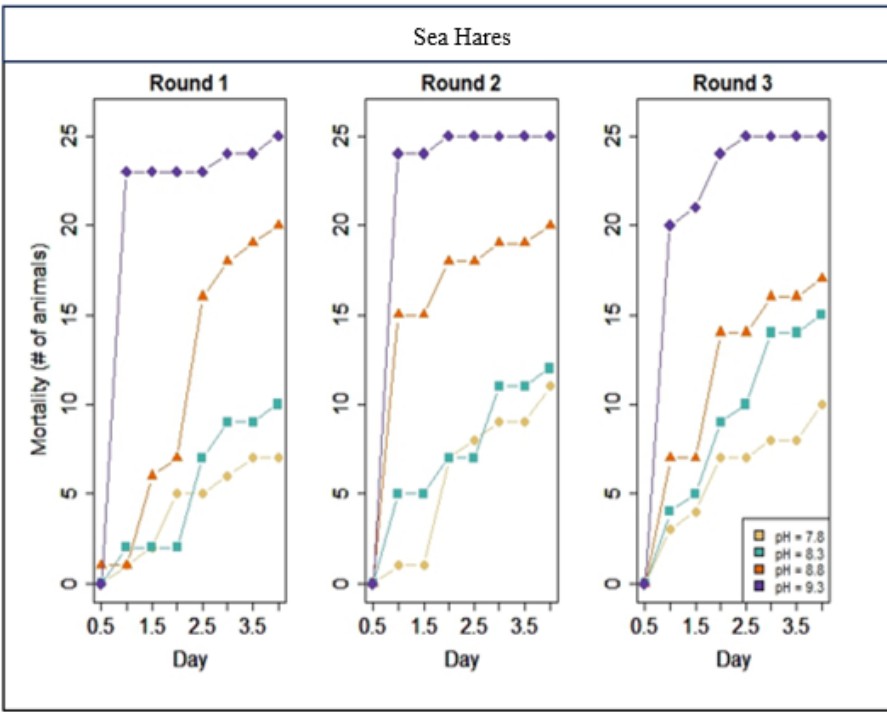



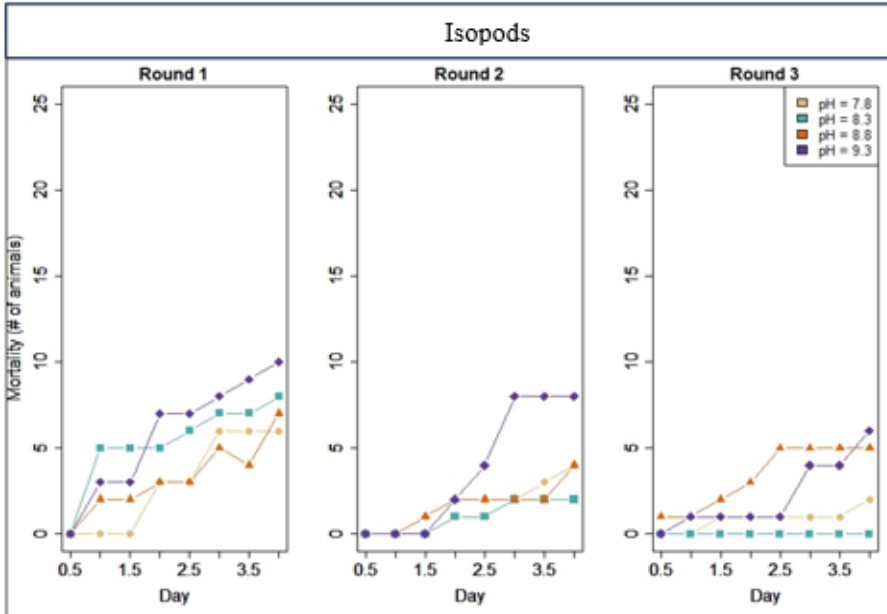

**Figure 4. Mortality of sea hares (top) and isopods (bottom) is shown over the four-day period for each of the three rounds of the**
**experiment and for each treatment group. Yellow circles indicate pH ≈ 7.8, blue squares indicate pH ≈ 8.3, orange triangles indicate**
**pH ≈ 8.8, and purple diamonds indicate pH ≈ 9.3.**

**Table 2. Tukey HSD matrices for sea hare and isopod mortality comparisons between treatments. Stars (\*) indicate a significant p-value of p < 0.05.**

| Sea Hare Mortality | | | | Isopod Mortality | | | |
|---|---|---|---|---|---|---|---|
| Round 1 | | | | Round 1 | | | |
| | Low | Medium | High | | Low | Medium | High |
| Control | $p = 0.98$ | $p = 0.15$ | $p < 0.001$* | Control | $p = 0.33$ | $p = 0.99$ | $p = 0.18$ |
| Low | | $p = 0.27$ | $p < 0.001$* | Low | | $p = 0.43$ | $p = 0.98$ |
| Medium | | | $p = 0.02$* | Medium | | | $p = 0.25$ |
| Round 2 | | | | Round 2 | | | |
| Control | $p = 0.96$ | $p = 0.02$* | $p < 0.001$* | Control | $p = 0.90$ | $p = 0.99$ | $p = 0.27$ |
| Low | | $p = 0.06$ | $p < 0.001$* | Low | | $p = 0.94$ | $p = 0.07$ |
| Medium | | | $p = 0.22$ | Medium | | | $p = 0.22$ |
| Round 3 | | | | Round 3 | | | |
| Control | $p = 0.76$ | $p = 0.30$ | $p < 0.001$* | Control | $p = 0.62$ | $p = 0.008$* | $p = 0.25$ |
| Low | | $p = 0.85$ | $p = 0.003$* | Low | | $p < 0.001$* | $p = 0.02$* |
| Medium | | | $p < 0.001$* | Medium | | | $p = 0.42$ |




Averaged over the course of all three rounds, a clear trend of increasing mortality correlating with increasing pH can be observed for sea hares (Fig. 5 top). The low treatment group displayed the most variation in mortality over the three rounds, and the high treatment group showed the least variation in mortality due to each round resulting in 100% mortality. The average mortality was not significantly higher in the low treatment group than the control group but was significantly higher for both

medium and high treatment groups (Fig. 5 top).

While there appears to be a slight increasing trend in the average mortality of isopods with each treatment group, the overlap between treatment groups is considerable, and shows high levels of variation in results between the three rounds of experiments. Average isopod mortality was not significantly different from the control group to any of the treatment levels (Fig. 5 bottom).


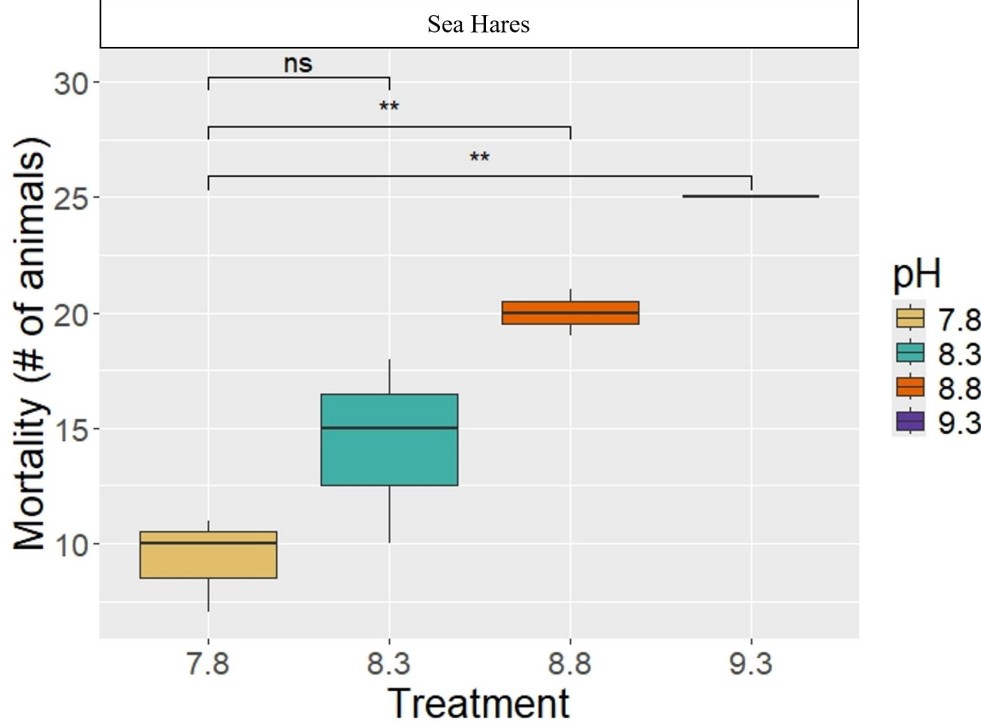



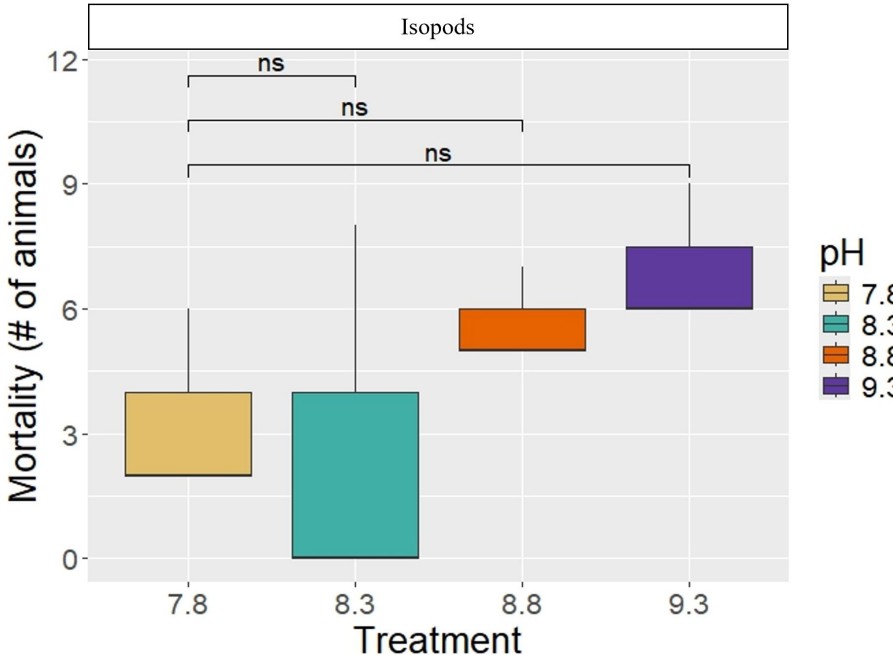


**Figure 5. Mortality of sea hares (top) and isopods (bottom) in each treatment group averaged over the three rounds of the experiment. T-test p-values are presented, ns indicates non-significance, ** indicates a significant p-value < 0.05.**

Multiple linear regression models for sea hare mortality data were created to determine if mortality was based only on changes
in pH, or if other environmental variables could have contributed, such as DO, salinity, and temperature. Fitting these models
and using the drop 1 function, pH was the only variable to have significant p-value in any of the models (Table 3). The same
models were made for the isopods; however, upon analysis none of the four variables had any significant effect on mortality
(no p-value < 0.05) in any of the models, so the data was not presented.

**Table 3. Multiple linear regression models for sea hare mortality p-value comparison using drop 1 function. Stars (*) indicates a significant p-value of p < 0.05.**

| Model | Variables | p value |
|---|---|---|
| death~ph + do + temp + salinity | pH | 0.03693* |
| | DO | 0.47927 |
| | Temperature | 0.91881 |
| | Salinity | 0.55388 |
| death ~ ph + do + salinity | pH | 0.03604* |
| | DO | 0.445 |
| | Salinity | 0.45183 |
| death ~ ph + do | pH | 0.02908* |
| | DO | 0.78374 |



### 3.3 Animal Lethal Concentration (LC$_{50}$)

On average, 50% mortality (LC$_{50}$) of the sea hares was observed after the first water refresh (between day 0.5 and day 1) in the high treatment group (pH 9.3) and after 1.5 days (between day 1 and day 2.5) in the medium treatment group (pH 8.8) (Fig. 6 top). In the low treatment group (pH 8.3), LC$_{50}$ was reached in only one of the three rounds of experiment, at day 3, and never reached in the control group (pH 7.8) (Fig. 6 top).

For the isopods, LC$_{50}$ was never reached in any treatment group, in any round of the experiment (Fig. 6 bottom). While some mortality was observed in every treatment group, isopod moulting and reproduction was observed throughout the experiment.

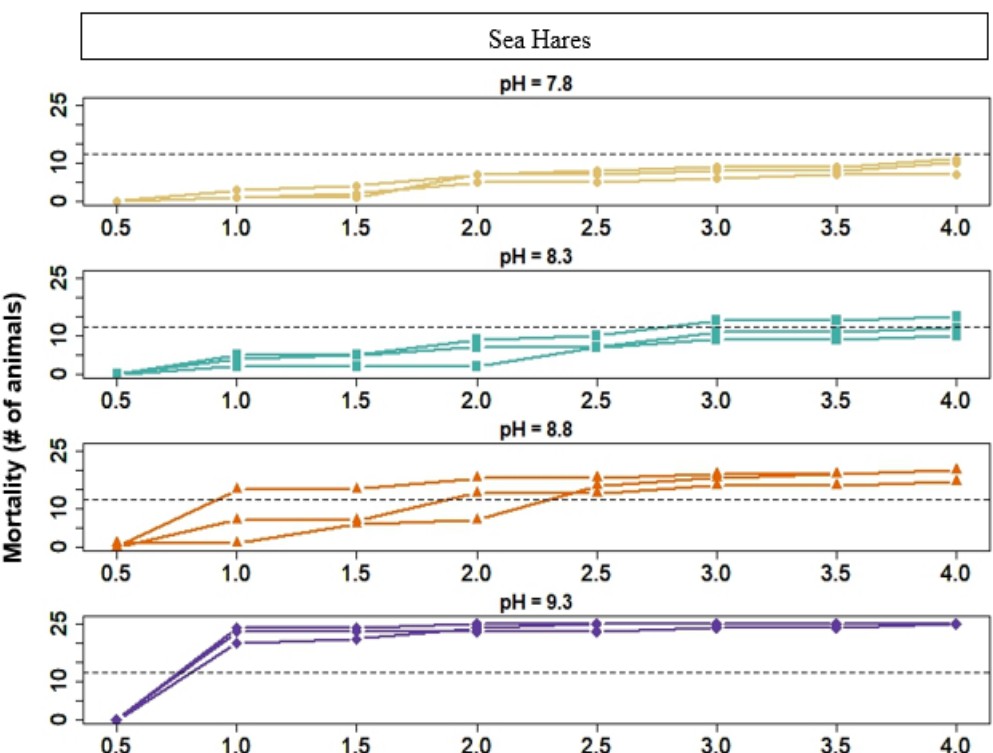





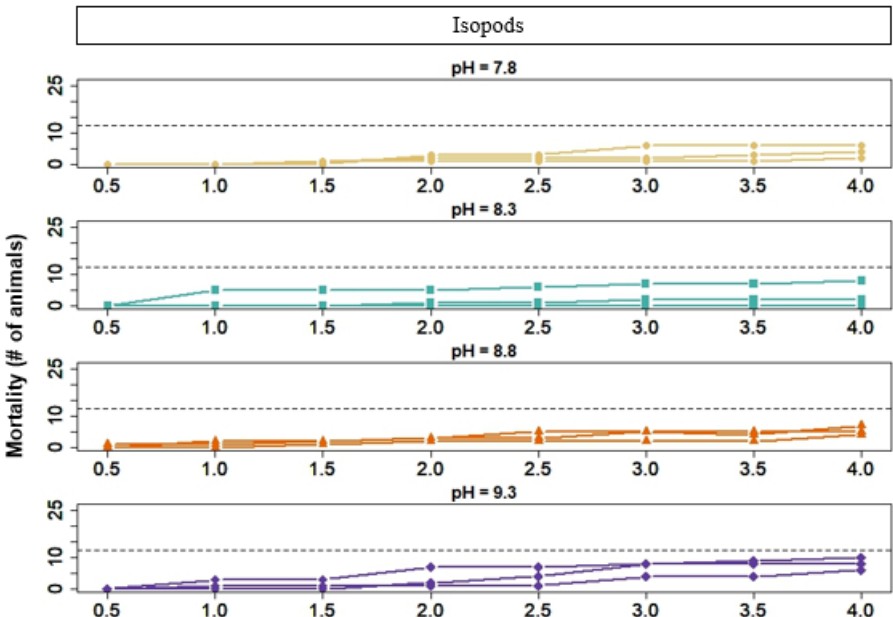

 **Figure 6. LC$_{50}$ plots for sea hare (top) and isopod (bottom) mortality at each treatment group. 50% mortality is indicated by the dashed line.**

## 4 Discussion

### 4.1 Biological response

The two eelgrass epifauna invertebrates investigated in this study, isopods and sea hares, responded differently to seawater
treated with NaOH. In increasingly alkaline environments with increasing pH, sea hares exhibited higher mortality in a shorter
amount of time than isopods, with over 50% mortality occurring in pH 8.3 within two days and reaching 100% mortality after
one day in pH 9.3. Isopods did not reach 50% mortality in any pH treatment.

In addition to showing higher survivorship, isopods exhibited growth and reproductive behaviors during the experiments. The
process of molting for eelgrass isopods promotes growth, as well as provides the animals with an opportunity to reproduce
(Kuris et al., 2007; Sadro, 2001). The isopods in the experiment exhibited molting in the control group and all treatment groups
in all three rounds of the experiment, indicating that growth and reproduction still occurred in highly alkaline waters. Because
the variation in mortality between control and treatment groups was small, we suggest that the mortality observed was likely
not due to variations in pH. During the experiment, isopods were observed cannibalizing each other in control and all treatment
groups. It is possible that the mortality we observed during the experiment could have been due to lack of food rather than
changes in pH. A previous study showed *Idotea resecata* as being the fastest daily consumers of eelgrass among six common
mesograzer species (Best and Stachowicz, 2012), suggesting they may not handle four days of fasting well. A previous study



investigating resilience to ocean acidification found opposing results, in that sea hare mortality was low while isopod mortality was high, but isopod mortality was likely not due to pH (Hughes et al., 2017). However, there is limited information on sea hare and eelgrass isopod biology, and this warrants further studies.

Few previous studies have investigated the effect of increased alkalinity and pH on various marine organisms. Cripps et al. (2013) investigated the response of European green crabs to increased alkalinity (through the introduction of calcium hydroxide) and found that female crabs were more susceptible to increases in pH affecting their physiology, though no mortality in any control or treatment group was observed. Shellfish species (blue mussel, eastern oyster, and bay scallop) were assessed for their behavioral response to increased alkalinity (through the introduction of calcium hydroxide). All three species

exhibited behavioral stress, but responses were short lived, and recovery occurred after treatment was halted (Comeau et al., 2017).

Results from these previous studies and the present experiments indicate that different species will likely have different responses to increased pH, depending on their physiological vulnerabilities. Investigation of the potential vulnerabilities of local marine species can help inform decision making in the mCDR planning process regarding the placement of highly

alkaline seawater outflow pipes in nearshore environments. This will also allow scientists to perform targeted monitoring on specific species that might be more sensitive during this process within the mixing zone of mCDR projects.

Additionally, the dependence of mortality on the duration of exposure is critical to the practical application of these results. In practice, the maximum pH resulting from an OAE intervention will be observed at the outfall or point of dispersal. The pH will decrease rapidly with distance through the mixing zone until the alkaline plume has diluted enough to where it will be

indistinguishable from natural variation in the open environment (Ho et al., 2023; Wang et al., 2023). If the species are mobile over an area larger than the mixing zone radius, they will only experience the mixing zone maximum pH for some period of time. However, immobile species, or species with small mobility ranges such as sea hares and isopods, located in the mixing zone will experience it continuously. Understanding the potential exposure of OAE projects on marine organisms will likely involve a combination of near-field dilution modeling of the release of alkalinity into seawater and in-situ sensing for pH

changes within the mixing zone. Studies investigating the impact of increasing alkalinity and pH on specific species should take into account the natural chemical variations experienced before an OAE perturbation, the range of chemical changes during an OAE perturbation based on the expected dilution of alkalinity and pH in time and space at varying distances from an outfall, and the potential for acclimatization to increased alkalinity and pH as OAE projects scale up from pilot experiments.

### 4.2 Limitations and Future Recommendations

To avoid air exchange causing the pH to drop below the target level, we needed to secure our experiment jars with airtight lids. This also prevented any animals from escaping the experiment. While this kept the pH within 0.1 of our target pH, it did cause some lowering of dissolved oxygen levels between water changes (on average, sea hares: -2.67 mg/L, isopods: - 0.57 mg/L). This was mitigated by refreshing the water twice daily, however in an ideal experiment, we would have a flow-through



system. A flow-through system would have a continuous flow of the treated water so the organisms would have ample oxygen
and the water would not need to be manually refreshed, reducing the need for physically handling the organisms. Decreasing
handling of organisms during the experiment might reduce unnecessary stress on sensitive species. Due to funding and
wastewater safety considerations, a flow-through system was not feasible at the time of this study.

It is also worth noting that during the initial treatment of seawater with NaOH, precipitation was visually observed but rapidly
dissipated upon mixing, indicating that this was likely $Mg(OH)_2$ (Ringham et al., 2024). No additional precipitation was
observed after incubation or before refreshing water throughout the experiments, even for the high pH treatment (9.3),
suggesting that these experiments did not surpass thresholds for runaway $CaCO_3$ precipitation, in which more alkalinity would
have been removed by precipitation than was initially added by the alkalinity treatment (Moras et al., 2022; Hartmann et al.,
2023; Suitner et al., 2023). Determination of precipitation thresholds is a major area of research for OAE because $CaCO_3$
precipitation can reverse the intended effect of OAE by removing alkalinity from the surface ocean and releasing $CO_2$ to the
water column. In addition, the increased turbidity resulting from precipitation may impact photosynthesis and predator-prey
interactions in the natural environment. Understanding connections between changes in carbonate chemistry and biological
activity is crucial for characterizing potential interactions between OAE and the biota in the receiving environment.
Experiments like those presented here will provide important baseline information for permitting OAE, particularly in coastal
waters where shallow well-mixed waters interact strongly with benthic biological communities that host organisms like sea
hares and isopods that are critical to marine food webs.

## 5. Conclusion

Understanding the biological implications of mCDR methods is an important consideration as this field expands. This current
study focused on the species level effects of locally important species in the Pacific Northwest. Future research should focus
on additional commercially, culturally, or ecologically important species. This work should be scaled up to understand the
effects of alkalinity enhancement on an ecosystem level using mesocosm studies, allowing for a more realistic depiction of a
natural ecosystem while still keeping the treatment confined and allow for both benthic and pelagic species or ecosystem
studies (Riebesell et al. 2023). Future research could leverage lessons learned from laboratory experiments and mesocosm
studies to eventually scale up to small scale field studies in sites under consideration for enhanced alkalinity approaches
(Cyronak et al. 2023).

**Data Availability**

Raw data will be made available by the authors.



**Author Contributions**

Kristin Jones:  Investigation, Formal analysis, Visualization, Writing - original draft preparation, Writing - review & editing

Lenaïg G. Hemery: Conceptualization, Methodology, Supervision, Writing - original draft preparation, Writing - review & editing

Nicholas D. Ward: Funding acquisition, Methodology, Project administration, Writing - original draft preparation, Writing - review & editing

Peter J. Regier: Formal analysis, Visualization, Writing - original draft preparation, Writing - review & editing

Mallory C. Ringham: Writing - original draft preparation, Writing - review & editing

Matthew D. Eisaman - Writing - original draft preparation, Writing - review & editing

**Competing Interests**

Matthew D. Eisaman is Co-Founder and Chief Scientific Advisor at Ebb Carbon, Inc.

**Acknowledgments**

This study was led by Pacific Northwest National Laboratory which is operated for the U.S. Department of Energy by Battelle Memorial Institute under contract DE-AC05-76RL01830. Funding was provided by a multi-agency award funded by the NOAA National Oceanographic Partnership Program, U.S. Department of Energy Water Power and Technology Office (WPTO), and Climate Works as part of the *Electrochemical Acid Sequestration to Ease Ocean Acidification (EASE-OA)* project, led by Brendan Carter (UW-PMEL), who we thank for his assistance throughout. The WPTO funded portion of the EASE-OA project was a sub-task of the Oceans for Climate AOP led by Chinmayee Subban.  We thank Ebb Carbon team members and especially Tyson Minck for helping us understand the electrochemical OAE process and for assisting with the experimental setup. The team also acknowledges former PNNL interns Kira Burch and Grace Weber, and PNNL researcher Tristen Meyers for their help running the experiments, and PNNL staff Jakob Bueche for assisting with the experimental setup and Ioana Bociu and Corey O'Donnell for assisting with permitting and compliance.



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
