# Peer review of "Biological response of eelgrass epifauna, Taylor's Sea hare (*Phyllaplysia taylori*) and eelgrass isopod (*Idotea resecata*), to elevated ocean alkalinity"

_EGUsphere, 2024_

## Author Comment (AC1)

We greatly thank the reviewer for their time reading our manuscript and providing such constructive feedback. Below, we answer their questions and detail our plan for addressing their comments and concerns in the revised version. Author responses begin with R:

Reviewer #2

The manuscript '*Biological response of eelgrass epifauna, Taylor's Sea hare (Phyllaplysia taylori) and eelgrass isopod (Idotea resecata), to elevated ocean alkalinity*' by Jones et al. is an interesting study on the topic of ecological safety of one of the promising marine carbon dioxide removal (mCDR) approaches Ocean Alkalinity Enhancement (OAE). This study aimed to analyze the survivability of two ecologically significant eelgrass epifauna to OAE-induced seawater chemistry changes, specifically increased pH. They observed the epifauna isopoda species *Idotea resecata* to be more resilient to elevated pH than the studied sea hare species *Phyllaplysia taylori*.

Although this incubation study might contribute to the baseline information in understanding OAE impacts on marine ecosystems, the manuscript contains substantial flaws making it unsuitable for publication in its current form. The main reasons for rejection are as follows:

1.  Missing total alkalinity measurements and complete focus on pH alone

R: As noted in our response to reviewer 1, we will include calculations of carbonate speciation and alkalinity based on pH and $pCO_2$ in the revised manuscript. Considering we used NaOH, changes in pH are directly related to changes in alkalinity. Also, while our lab has alkalinity and DIC measurements capabilities, it was not within the scope of this project to analyze the many hundreds of bottle samples that would have needed to be collected to cover all the replicates and water changes. At the time, $pCO_2$ measurements were much higher throughput for our team.

2.      Lack of proper statistical analysis

R: These experimental data are not sufficient to conduct more sophisticated statistical analyses than the ANOVAs, Tukey tests, and multiple linear regressions presented here.

3.      There is no point in plotting the three experimental rounds separately

R: The figures were revised to plot all three rounds at once, as averages with standard deviations, and the figures with three rounds separately will be included in supplementary material.

4.      Short and limited discussion

R: Thanks to the constructive comments of both reviewers, we have greater material to now expand the discussion in the revised version: e.g., LOEC and NOEC in addition to LC50, applicability of ecotox experiments to OAE effects, direct application to OAE field trials (continuous exposure in mixing zone).

**Specific comments**

33      The author should concise the overall OA impact on marine life. They could then consider adding specific information on OA's impact on the eelgrass ecosystem or the influence on their epifauna.

R: Based on available scientific literature, information about OA impacts on eelgrass ecosystems and our two examined species (or related species) will be added in the revised version. We have added a brief overview of OA impacts on eelgrass ecosystems and the specific species used in this study to the

introduction at lines 115-125, which now reads: "Research on ocean acidification effects for the two species examined is limited, however one study (Hughes et al. 2017) investigated OA effects on *P. taylori* and *I. resecata* and found a negative quadratic relationship between change in biomass and pH. This study also observed low sea hare mortality and high isopod mortality in response to OA (Hughes et al 2017). Studies investigating OA effects on other species of sea hares and isopods can be used to provide context. When exposed to OA conditions, the California sea hare, *Aplysia californica*, experienced altered behavior and acid-base regulation (Zlatkin and Heuer 2019, Zlatkin et al. 2020). When exposed to OA conditions, behavior of sea hare *Stylocheilus striatus* was altered, showing reduced speed, and foraging success, and exhibiting increased metabolic demand (Horwitz et al 2020). Another species of isopod, *Idotea balthica*, when exposed to OA conditions, exhibited 100% mortality in high $pCO_2$ conditions (Wood et al. 2014). Eelgrass mesograzer species sensitivity to shifts in environmental changes, such as pH, salinity, and temperature are likely to affect their feeding on epiphytes, which can lead to indirect effects on the growth and productivity of the eelgrass ecosystem (Hughes et al., 2017; Tanner et al., 2019)."

39    The first and second paragraphs do not naturally fit together. The author starts by introducing Ocean acidification (OA) and its impact on marine life, then abruptly shifts to mCDR and OAE. To bridge the gap, the author should add a line stating that OAE could counteract OA, making the connection between these topics more obvious.

R: In thinking about how to better fit these two paragraphs together, we decided to swap them and start the revised introduction by the need to reduce atmospheric $CO_2$ and OAE as a potential approach, followed by the description of ocean acidification, which also fits better with the following paragraph describing the biological effects of OA and OAE.

42    Alkalinity enhancement is more than that, this is one possible method. This should be clarified.

R: Clarification and proper references will be added in the revised version. Lines 32-45 describe one specific method of OAE that involves the electrochemical removal of acid from seawater, resulting in increased TA and pH in water returned to the surface ocean. This study was designed to understand the impact of the change in seawater chemistry in response to this specific approach to OAE but may not be applicable to all methods of OAE. Because other approaches to OAE (or mCDR more broadly) are not within scope of this study, we limited discussion in the first version of our introduction– however we agree that this may not be clear to all readers. To disambiguate, we will revise this paragraph to the following: *"Natural carbon sinks on land and in the ocean help to reduce atmospheric $CO_2$ but are not keeping pace with increasing anthropogenic emissions, prompting research efforts to explore methods of enhancing the ocean's natural carbon sink through marine carbon dioxide removal (mCDR) (National Academies of Sciences, Engineering, and Medicine, 2022). One method of mCDR, ocean alkalinity enhancement (OAE), aims to store atmospheric $CO_2$ in seawater in the dissolved bicarbonate phase in response to a disequilibrium in $pCO_2$ across the air-sea boundary of the surface ocean, induced by a change in seawater alkalinity (Cross et al., 2023; Oschlies et al, 2023). OAE approaches include the addition of natural or industrial alkaline materials to ocean or coastal environments (Feng et al., 2017; Köhler, Hartmann, and Wolf-Gladrow, 2010; Monserrat et al., 2018; Rigopoulos et al., 2018; Harvey, 2008; Ilyina et al., 2013; Kheshgi, 1995; La Plante, 2023; Moras et al., 2022; Nduagu, 2012; Rau, 2008; Renforth and Henderson, 2017; Shaw, 2022), and electrochemical approaches that process salt (e.g., sodium chloride NaCl) to generate aqueous acid (hydrochloric acid HCl), which is removed from the system, and base (sodium hydroxide NaOH), which is mixed with the seawater stream and returned to the ocean, thus enhancing the alkalinity of the surrounding water (de Lannoy et al., 2018; Eisaman et al.,*

*2018; Eisaman et al., 2023; Lu et al., 2022; Ringham et al., 2024; Tyka et al., 2022; Wang et al., 2023; Eisaman, 2024). The increased surface alkalinity drives additional ocean uptake of atmospheric $CO_2$, which is ultimately stored in seawater as dissolved bicarbonate (Cross et al., 2023, Eisaman et al., 2023, Ringham et al., 2024). While OAE could be a promising avenue for reducing atmospheric $CO_2$, the biological response of marine organisms and impact on ecosystems of locally increasing pH and alkalinity remains largely unknown."*

55      acclimate to what?

R: Text has been added  to line 74 that states "to local environmental changes".

72      Scientific name of eelgrass should be provided.

R: "*Zostera marina*" has been added to the revised manuscript in line 91.

80      Specify what is synergistic with what to avoid ambiguity.

R: Text has been rephrased to "potential impacts on eelgrass epifauna under either acidic or alkaline treatments remain unknown."

84      The authors mentioned why these two epifauna species are important for the eelgrass ecosystem. They only connected the importance of the isopoda species with higher trophic levels as prey for Pacific salmon. Could the author mention similar for the sea hare species to support the importance of studying this particular species of gastropod? The authors might consider adding information if this species is prey for fish, crustaceans, or sea birds. Right now, it seems this sea hare species is only contributing to reducing the epiphyte load from the eelgrass blade.

R: Based on available literature, it is unclear what animals may predate on this specific species of sea hare. Its main ecological role is as an epiphyte grazer in eelgrass beds (Tanner,  2018). This sentence is now added to the revised manuscript at line 108 and states: "As grazers, Taylor's sea hares are main contributors to reducing the epiphyte load from eelgrass blades (Tanner 2008)."

89      In line 85, it is mentioned that *Idotea* sp. feeds on eelgrass blades, while here it is stated they feed on epiphytes. This inconsistency needs clarification. If line 85 means that they physically sit on the blades then the sentence is not well formulated.

R: Text has been rewritten in the revised manuscript at line 104 and now states "Isopods crawl on the eelgrass blades and feed on epiphytic diatoms."

96      Why was the study conducted in an environment with such a large natural pH variation? The organisms might have already acclimated to large pH variations.

R: While pH in the Puget Sound can vary from 6.5 to 8.5, in the near vicinity of the laboratory's water intake it is more between 7.6 and 8.2 (Cotter et al. 2022 and unpublished data). As per Reviewer 1's suggestion, this information has been added to the revised version. In addition, the experimental treatments targeted pH higher than the local variability (8.3, 8.8 and 9.3). These nearshore ecosystems are also likely targets for OAE projects in the near term considering the substantially more challenging

logistics of developing an offshore facility. For example, our lab currently has a pilot electrodialysis OAE system installed and generating base and will likely perform field trials in the future. The data produced by these experiments are essential for assessing the potential impact of such trials and deployments, and will help pave the way for responsible permitting and regulation of OAE deployments.

110    I found it difficult to understand the experimental setup, specifically the details regarding the number of organisms and replications. I would recommend revising the methods section to help readers easily understand how the experiments were performed.

R: The description of the experimental setup in the methods section has been clarified in the revised version to help readers understand the study. Especially, a summary statement was added to clarify the numbers of jars, animals, and controls at lines 197-201: "Each three rounds of the experiments for both sea hares and isopods comprised a total of 24 jars: five at control pH (no NaOH added) with five animals each for a total of 25 animals, one chemical control (without animals) at control pH, five at each of the three pH treatments (NaOH added) with five animals each for a total of 25 animals per treatment, and one chemical control (without animals) for each of the three pH treatments (NaOH added)."

118    Why do animals need nutrients?

R: All living organisms need nutrients for survival, growth, etc. However, the sentence was rewritten as follows to avoid confusion "continuously flowing raw seawater from Sequim Bay to provide adequate water flow to the organisms".

135    The manuscript highlighted the pH increase as the seawater chemistry change of OAE. If this study is about biological responses to elevated ocean alkalinity, it should at least mention the initial total alkalinity (TA) value and the TA values after NaOH addition. Also, I didn't understand the reason for targeting a specific pH instead of targeting alkalinity for the treatment manipulation.

R: As stated in previous responses, a full description of seawater carbonate chemistry will be provided in the revisions to clarify the induced changes in pH, TA, and the addition of NaOH. There are 2 major reasons to focus on the targeted pH's instead of alkalinity: one is logistical- measurement of pH via probe is a faster and far more accessible analysis than a laboratory titration of TA, allowing for rapid refreshment of treated water to animals during these experiments, though we note that the advantage of speed in pH probe measurements is tempered by the error inherent in these measurements. In this experiment, TA changes can be calculated from the known NaOH addition, given a constant concentration of a commercial NaOH source carefully dosed into seawater to raise the pH to a given target level. While direct measurements of TA before and after NaOH additions would be ideal, the tradeoff between labor and the potential error in dosing TA is significant. The second reason to focus on pH is in the application of this type of study to decision making in OAE field trials– coastal OAE trials in the US may be subject to regulation by Clean Water Act NPDES permits, which frequently include the designation of a mixing zone within which water quality standards may be exceeded locally and by the edge of which an effluent must be sufficiently diluted to be indistinguishable from natural variation. NPDES permits typically include specification of water quality parameters including pH, TSS, temperature, turbidity– and not TA. As already stated in the introduction, "pH shifts can affect physiology of aquatic organisms by disrupting acid-base regulation essential for cellular function, can inhibit fixation and respiration of $CO_2$, and reduce nutrient uptake (Tresguerres et al. 2020, Tresguerres et al. 2023)". These topics will be more closely tied together with the addition of complete carbonate chemistry tables in the revised manuscript.

150   Fig 1 needs to be improved with additional information to simplify the complex setup.

R: Figure 1 has been improved for the revised version (see above). The number of acclimation tanks does not relate to the number of treatments. As explained L141-142, 5 individuals were randomly picked from each acclimation tank and moved into a jar at the start of each round of experiment. This will be clarified in the revised version and represented in the revised figure. In addition, the differences between the types of controls (non-treated water with animals, non-treated water without animals, and treated water without animals) will be clarified in the revised text and better represented in the revised figure. The figure caption now reads "Figure 1. Experimental setup of laboratory water table, including the 1-liter glass jars (colored circles) used in the experiment and the acclimation tanks (blue rectangles). The white circles indicate control jars with pH ≈ 7.8 (no NaOH added), yellow circles indicate low treatment with pH ≈ 8.3, orange circles indicate medium treatment with pH ≈ 8.8, and red circles indicate high treatment with pH ≈ 9.3. The dashed circles with "C" indicate the chemical control jars in which only treated, or control, seawater was added without the presence of sea hares or isopods. Animals were distributed randomly from the acclimation tanks to the experiment jars, as indicated by the black arrows. The circles "W" indicates the water input, the circle with "D" indicates the drain on the water table, and the blue arrows indicate the water flow along the table."

156   rather stressful for the animals, or not?

R: The water refresh process was handled as carefully as possible to create as little stress for the animals as possible. Leaving the water without an exchange would have been more stressful considering O2 levels would decline among other changes. Changes in DO levels in between water refreshes will now be available in the supplementary material table, allowing the reader to understand how quickly the jars would have become anoxic without refreshing the water twice daily.

161   I would not classify reproduction as unusual behavior.

R: This sentence has been rewritten in the revised manuscript to say "noticeable behavior".

166   Why there are ranges for all physio-chemical paraments?

R: This table is intended to give context for the basic water quality conditions for the experiments. Information such as differences (or lack thereof) of temperature, salinity, and oxygen are important for confirming that there were or weren't confounding factors to consider. Based on Table 3, we show that mortality is most closely related to pH rather than variability in these other parameters. While not terribly exciting, this data provides important context.

180   This is a pity. At least in the starting water total alkalinity should have been measured, and also at least a few times in the treatments. We have no idea about total alkalinity now, and whether there were risks of precipitation (no indication either whether this was observed).

R: As explained above, total alkalinity was not measured directly but was calculated from the measured pH-pCO$_2$ data (albeit at poorer resolution and uncertainty than direct TA measurements themselves), along with the known addition of NaOH. This data will be provided in the revised manuscript. No precipitation was observed in the treatment water aside from some initial precipitation (assumed to be brucite (Mg(OH)$_2$) when adding base that dissolved after gently mixing.

186     mortality was noted every day, how was this dealt with?

R: As noted L123-124 & 161-162, any casualties were removed from the tanks/jars during the twice-daily checks.

200     The $pCO_2$ values of the controls are very high, why?

R: Text has been added to the discussion in the revised manuscript at line 474-480 and states: "In the experimental setup, the mason jars were closed tightly and contained active microbial communities from the raw seawater. The PNNL wet laboratory has normal indoor lighting, but we did not subject the jars to grow lights or nighttime lighting, and does not have natural lighting, therefore algal production inside the jars likely slowed or stopped, while microbial respiration continued. This respiration would have resulted in $CO_2$ production and $O_2$ consumption, leading to high $PCO_2$ measurements in the chemical control jars. In the jars with organisms present, $CO_2$ production was a result of microbial and animal respiration, which could lead to declines in pH." Likewise, the animals were actively respiring until dead, producing elevated $pCO_2$ and lowering pH. This is why we refreshed the water twice per day to maintain pH near our target treatments. When sealing water in a closed system without natural light, the outcome is generally net heterotrophy and $CO_2$ production.

245     Presumably the $CO_2$ in the animal containers comes from respiration, but if the animals played such a large role in the $pCO_2$, how relevant was the treatment?

R: $CO_2$ generated in the jars was a combination of microbial and animal respiration. As $CO_2$ is produced, pH is expected to decline, which is why we refreshed the water twice per day to maintain our target pH as close as possible to the desired levels. It is essentially impossible to culture organisms with no changes in $CO_2$ in a closed system, without using a chemostat system that constantly refreshes the water, which was not in the scope of this experiment.

245     In Fig 2, the y-axis title is not included in all plots. If the authors want to show the y-axis title for the left plot only, then they should do the same for all figures. The y-axis title is labelled as $CO_2$, but it displays the $pCO_2$ data. Why are the symbols for pH 7.8 and 8.3 different in rounds 1 to 2 and 3?

R: Thank you for your comment, attached are the new figures with corrected symbols and added y axis labels (sea hares no organisms present on top, sea hares organisms present on bottom). See Figure 2 below. The same revisions have been done for the isopod $CO_2$ plots.

[Figure]

Figure 2: pCO₂ (ppm) measured in sea hare jars, before and after twice daily water changes for both jars without organisms (top), and jars with organisms (bottom). Closed symbol = start of a segment using refreshed water, open symbol = end of a segment before a water change. Yellow circles indicate pH ≈ 7.8, blue triangles indicate pH ≈ 8.3, orange squares indicate pH ≈ 8.8, and purple diamonds indicate pH ≈ 9.3.

322    Here and in other places, I do not understand why the analysis was not done using all rounds in a more sophisticated analysis, it is not so relevant that in round 1 this happened and in round 3 something

else, unless you can explain. The general picture is more important, though. So, I would not present the rounds separately, but as a composite picture

R: Thank you for this suggestion. We agree that the general picture is more important, and while we originally separated our dataset by round of experiment, we have now grouped our dataset to make the general trends in our dataset clearer. The description of the results will be rewritten as suggested.

345    In Fig 4 authors may want to consider adjusting the maximum limit of the y-axis to 10.

R: Thank you for your suggestion, however the total number of animals per treatment was 25, therefore the maximum mortality possible. Keeping the axis at 25 allows an easy comparison of mortality between the two species. The figure was edited, and the caption will be clarified in the revised manuscript to represent this, see 4 below.

[Figure]

Figure 4. Average mortality of sea hares (left) and isopods (right) is shown over the four-day period over the three rounds of the experiment and for each treatment group. Yellow circles indicate pH ≈ 7.8, blue circles indicate pH ≈ 8.3, orange circles indicate pH ≈ 8.8, and purple circles indicate pH ≈ 9.3.

345    Table 2, again. It is not so interesting to see the rounds separately. The reader would like an overall picture. If you want to present put Figs 1-4 in the supplementary material.

R: New figures have been created showing the overall picture of mortality averaged over all three rounds (see Figure 4 above). The figures separated out into each round will be added to supplementary material. Table 2 will be remade to show the overall picture.

375     I do not understand the word t-test in the caption of Fig 5. What was exactly tested, I hope not everything is against everything, using t-tests, would be incorrect as it would involve multiple testing with inflated type 2 error. The data needs to be analyzed better.

R: Thank you for this comment: We used pairwise Wilcoxon tests to compare each pH treatment to the control pH (7.8). With this test, we get p-values for each of the following treatment combinations: 7.8 vs 8.3, 7.8 vs 8.8, and 7.8 vs 9.3. Because we have the same number of samples per treatment, we are comparing equal sample sizes for all combinations. In addition, we are not comparing everything to everything, but rather conducting separate statistical tests comparing each of the combinations above as its own test. We conducted our tests in this manner since we wanted to understand if each treatment level was significantly different from our control (ambient pH). This has been clarified in the text at line 251-252 stating "Pairwise Wilcoxon tests were conducted to compare average mortality from each pH treatment (8.3, 8.8, and 9.3) to the control pH (7.8)."

375     The quality of the figures needs improvement. The picture quality of Figures 4 and 5 is much better than that of the other figures. Additionally, it's preferable to keep the font size of the axis labels, titles, and legend text consistent across all plots.

R: Thank you for your comment, this will be rectified in the revised figure.

379     Was there enough variation in any of these variables to warrant an inclusion in the model, probably not.

R: Our intention with this model was to examine if drivers other than our intended manipulation (pH) caused mortality. For example, did a lack of oxygen, difference in temperature, or change in salinity confound our results? The lack of correlation with parameters other than pH confirm that pH indeed was the cause of mortality differences across treatments assuming malnourishment equally affected each pH treatment. Variations in salinity, DO, and temperature are provided in Table 2.

390     x-axis title is missing in Fig 6. LC50 describes a concentration. I do not see a concentration in this figure.

R: The x-axis will be added at revision. You are correct, instead of LC50, the figures and captions will be revised to present and explain the exposure time to each treatment after which 50% mortality occurs. We will also present NOEC (no observed effect concentration) and LOEC (lowest observed effect concentration), using pH as a concentration and clarifying that in the revised text. Text has been added at lines 456-458 that states: "The lowest observed effect concentration (LOEC) for sea hares was pH of 8.8, and the no observed effect concentration (NOEC) for sea hares was pH of 8.3. For isopods, an LOEC and NOEC were not able to be determined because no mortality in any treatment group was statistically higher than mortality in the control group."

408     Do the observed reproduction and growth rates remain consistent across all pH levels?

R: Isopod molting (an indication of reproduction and growth) was observed in all pH treatment levels and control. This was observed as a behavior and was not measured as a rate. Growth rates were not measured.

412   Essentially your tests state that it is not, so there is no suggestion here it is the outcome of the study.

R: The text has been modified at lines 491-492 and states: " The variation in mortality between control and treatment groups was small and statistically insignificant, demonstrating that the mortality observed was likely not due to variations in pH."

414   It is also possible that growth and reproduction were done by those animals that had energy intake.

R: We will never know; it was not possible to individually recognize each individual in the jars to track them over the duration of each round.

414   The author conducted the study using starvation due to standards for acute toxicity tests with macroinvertebrates. The observed responses in the survivability of sea hares might have intensified due to starvation, leading to an overstretched result from an OAE perspective.

R: This is possible and will be clarified in the revised discussion. Text has been added to the limitations section of the manuscript at lines 535-537 and states: "According to ASTM 2000 standards for acute toxicity tests (ASTM, 2000), the experiments were conducted under starvation conditions. Due to this, the observed responses in the survivability of the sea hares might have been intensified by the starvation, leading to an overstretched result from an OAE perspective."

466   In the discussion, the author should include the pH ranges of the mentioned previous OAE studies.

R: Good suggestion, thank you. pH ranges from previous studies have been added to lines 503 and 506.

466   Overall the discussion is very short and lacks focus on the specific findings of this study. There is a significant lack of discussion regarding the high mortality observation in sea hares. Specifically, if the results are so contrasting with the OA study by Hughes et al. (2017), the authors should at least mention what may have influenced the outcomes in their study. The discussion part needed information on how pH imbalance might have impacted physiological functions in sea hares, which could have contributed to the high mortality. If there is no information regarding this particular species, then discuss the effect of pH change on other sea hares or other gastropod species.

R: Once introduction, methods, and results are revised, the discussion will be expanded to reflect the revised information, including suggestions made by the reviewer: e.g., LOEC and NOEC in addition to LC50, applicability of ecotox experiments to OAE effects, direct application to OAE field trials (continuous exposure in mixing zone).

---

## Author Comment (AC2)

We greatly thank the reviewer for their time reading our manuscript and providing such constructive feedback. Below, we answer their questions and detail our plan for addressing their comments and concerns in the revised version. Author responses begin with R:

Reviewer #1

This study examines the biological effect of NaOH on two invertebrate species (isopod and sea hare) in a 4-day long experiment testing three different pH levels. The study presents the results that are indicative of high sensitivity of both species under elevated pH (aligned with stronger OAE treatment). There are serious data missing on the carbonate chemistry and the overlap in conditions among the experimental treatments shows that the analyses have to be conducted in a different way (see the text below).

The **introduction** addressed OA and OAE and provides a rational for the OAE testing. However, it does provide an insufficient background on the OAE effects across different species. It also fails to provide OA effects (low pH) on the two examined species, which would give the background on their sensitivity. This is pertinent to more detailed explanation in lines 92-95.

R: Additional information pertaining to OA effects on our study species and related taxa has been added to the revised manuscript at lines 115-126, stating: "Research on OA effects for the two species examined is limited, however one study (Hughes et al. 2017) investigated ocean acidification effects on *P. taylori* and *I. resecata* and found a negative quadratic relationship between change in biomass and pH. This study also observed low sea hare mortality and high isopod mortality in response to ocean acidification (Hughes et al 2017). Studies investigating ocean acidification effects on other species of sea hares and isopods can be used to provide context. When exposed to ocean acidification conditions, the California sea hare, *Aplysia californica,* experienced altered behavior and acid-base regulation (Zlatkin and Heuer 2019, Zlatkin et al. 2020). When exposed to ocean acidification conditions, behavior of sea hare *Stylocheilus striatus* was altered, showing reduced speed and foraging success, and exhibiting increased metabolic demand (Horwitz et al 2020). Another species of isopod, *Idotea balthica*, when exposed to ocean acidification conditions, exhibited 100% mortality in high $pCO_2$ conditions (Wood et al. 2014). Effects of OAE on other macroinvertebrates is limited and effects on the two species examined is unknown." Unfortunately, there is very little scientific literature available on the effects of OAE on macroinvertebrates (we already leverage all those we could find in the third paragraph of the introduction), and none on the two species examined.

Importantly, these are the sub-tidal species inhabiting the eel grass habitat but no data on the pH variability in the Puget sound and specifically within the eelgrass ecosystem is provided. Such information is essential for us to understand the diel variability these species encounter, as well as the pH max- namely, if subjected to high pH conditions in situ in this temporal period (Aug-Sept), then we might be expecting some natural acclimation/adaptation capacity in these species, that might be impacting the experimental results. Also, are there any changes in the month-long Aug-Sept period?

R: The revised version will include greater information about pH variability in the Puget Sound/Sequim Bay and in eelgrass habitats: "Based on Cotter et al. (2022), pH within an eelgrass meadow at the entrance of Sequim Bay varied between 8.02 and 8.22 between low and high tide and was consistently higher than outside of the meadow. Unpublished data show tidal variation of the pH in the tidal channel of Sequim Bay ranging from 7.6 to 8.2." This information was added to the manuscript at lines 130-132 and will be discussed accordingly based on the study results.

**Methodology** is the section where most of the clarifications are needed to evaluate the correctness of the exp design.

It needs to be noted that while the invertebrates living in the eelgrass are exposed to pH variability, none of the experiments incorporated this pattern in the exposed.

R: The experiments were designed to determine the pH level, in the range of possible OAE releases, that would be lethal to eelgrass invertebrates if exposed to these conditions continuously. For example, a sessile organism living nearest the outfall of an OAE facility that continuously releases seawater with elevated alkalinity to a defined mixing zone could be subjected to these extreme perturbations in pH and alkalinity. Some OAE deployments are currently being planned under wastewater discharge regulations (eg, NPDES under the US Clean Water Act in the US Pacific Northwest) that allow for specific water quality exceedances within a defined mixing zone. While pH must return to ambient conditions by the edge of the mixing zone, a planned deployment could potentially override natural variability (e.g., aim for an elevated pH >=9 in the mixing zone) to optimize alkalinity delivery. For the purposes of understanding extreme exposure cases, varying the pH throughout the day in each treatment group would have confounded the results. We agree that varying pH throughout the day would be similar to what a biological community currently experiences or would experience at some distance from the outfall, -- and we note that for a realistic OAE pilot deployment, continuous 24/7 release of alkalinity to maintain a constant elevated pH signal is unlikely. However, the goal of this study was to understand what constant elevated pH did to organisms as a first step to designing experiments that more closely mimic conditions related to an OAE field deployment, which we now expand in the introduction. The objectives, at the end of the introduction at lines 139-142, was rewritten as "The objective of this study was to assess the biological responses of eelgrass epifauna (Taylor's sea hares and eelgrass isopods) to increased pH and alkalinity levels and determine the pH level, in the range of possible OAE releases, that would be lethal to eelgrass invertebrates if exposed to these conditions continuously, in order to inform future mCDR field trials and identify knowledge gaps pertaining to laboratory and field trials in the context of OAE."

In addition, it is not provided how much of the eelgrass is sufficient for their nutrition and if the added eelgrasses sufficed or not- could there be some malnutrition occurring in the experiments? How was the diatom concentration determined, what equal concentration distributed within all the experimental jars? Are these two specific invertebrates feeding on diatoms and what is the heir C intake from eelgrass vs diatom?

R: As specified L125-126, the invertebrates were not fed during the 4-day experiments, per the ASTM 2000 standards. They were provided eelgrass blades with diatom clumps only during the acclimation period, in large quantities and refreshed daily. Malnourishment is one likely cause of mortality in the non-pH adjusted controls. We note that ASTM 2000 standards do not accurately represent OAE conditions that will be experienced by organisms within the mixing zone. We have added the following text to clarify this point within the methods section at lines 171-172 "This means that malnourishment may lead to some mortality, in both treatment groups and non-pH adjusted controls". This has been acknowledged in the limitations section of the manuscript at lines 538-540 stating "According to ASTM 2000 standards for acute toxicity tests (ASTM, 2000), the experiments were conducted under starvation conditions. Due to this, the observed responses in the survivability of the sea hares might have been intensified by the starvation, leading to an overstretched result from an OAE perspective."

Provide pH variability data as this represents a baseline to what these species are exposed to- lines 112-113.

R: A table will be added to the supplementary material with the data measured at each water refresh: pH, DO, salinity, temperature, pCO$_2$, and NaOH added. In addition, a summary table (see below) will be added to the results section about changes in pH and pCO$_2$, with averaged measured pH, pCO$_2$, and calculated alkalinity, DIC, and aragonite saturation. This is not a final table, but a quick example summary table. Full tables of each water refresh will be made available in the revision in the supplementary materials.

| Organism | Time Point | organisms present | Mean Salinity | Mean Temp (°C) | Target pH$_{NB}$ | Measured pH Mean sd | pCO2 (ppm) mean sd | Alkalinity (µmol kg$^{-1}$) mean sd | DIC (µmol kg$^{-1}$) mean sd | omega aragonite mean sd |
|---|---|---|---|---|---|---|---|---|---|---|
| Sea Hares | initial | No | 30.16 | 13.74 | 7.8 | 7.81 ± 0.13 | 646.55 ± 158.63 | 1955.7 | 1870.5 | 1.12 |
| Sea Hares | initial | No | 30.16 | 13.74 | 8.3 | 8.26 ± 0.04 | 317.56 ± 59.15 | 2146.4 | 1870.5 | 2.98 |
| Sea Hares | initial | No | 30.16 | 13.74 | 8.8 | 8.73 ± 0.02 | 205.26 ± 78.83 | 2530.6 | 1870.5 | 7.34 |
| Sea Hares | initial | No | 30.16 | 13.74 | 9.3 | 9.22 ± 0.04 | 197.03 ± 94.69 | 3135 | 1870.5 | 14.86 |
| | | | | | | | | | | |
| Sea Hares | final | No | 30.28 | 13.45 | 7.8 | 7.75 ± 0.13 | 685.74 ± 120.11 | 1791.4 | 1727.1 | 0.9 |
| Sea Hares | final | No | 30.28 | 13.45 | 8.3 | 8.21 ± 0.09 | 308.4 ± 67.07 | 1961 | 1727.1 | 2.45 |
| Sea Hares | final | No | 30.28 | 13.45 | 8.8 | 8.54 ± 0.66 | 181.27 ± 69.89 | 2176.8 | 1727.1 | 4.77 |
| Sea Hares | final | No | 30.28 | 13.45 | 9.3 | 9.16 ± 0.07 | 203.94 ± 267.32 | 2840.5 | 1727.1 | 12.72 |
| | | | | | | | | | | |
| Sea Hares | final | Yes | 30.9 | 12.56 | 7.8 | 7.5 ± 0.18 | 1243.86 ± 413.63 | 1775.9 | 1774.4 | 0.51 |
| Sea Hares | final | Yes | 30.9 | 12.56 | 8.3 | 8.07 ± 0.13 | 442.96 ± 101.43 | 1944.8 | 1774.4 | 1.83 |
| Sea Hares | final | Yes | 30.9 | 12.56 | 8.8 | 8.6 ± 0.1 | 244.87 ± 95.24 | 2279.4 | 1774.4 | 5.37 |
| Sea Hares | final | Yes | 30.9 | 12.56 | 9.3 | 9.13 ± 0.09 | 186.73 ± 166.36 | 2828.6 | 1774.4 | 11.95 |
| | | | | | | | | | | |
| Isopods | initial | No | 32.24 | 13.04 | 7.8 | 7.74 ± 0.11 | 752.7 ± 133.55 | 1943.9 | 1875 | 0.97 |
| Isopods | initial | No | 32.24 | 13.04 | 8.3 | 8.25 ± 0.04 | 361.47 ± 75.9 | 2156.4 | 1875 | 2.96 |
| Isopods | initial | No | 32.24 | 13.04 | 8.8 | 8.73 ± 0.02 | 230.59 ± 95.36 | 2558 | 1875 | 7.43 |
| Isopods | initial | No | 32.24 | 13.04 | 9.3 | 9.23 ± 0.03 | 245.65 ± 130.9 | 3185.3 | 1875 | 15.11 |
| | | | | | | | | | | |
| Isopods | final | No | 32.29 | 12.57 | 7.8 | 7.69 ± 0.15 | 778.66 ± 182.55 | 1782.8 | 1731.4 | 0.79 |
| Isopods | final | No | 32.29 | 12.57 | 8.3 | 8.2 ± 0.07 | 356.56 ± 70.84 | 1968.2 | 1731.4 | 2.42 |
| Isopods | final | No | 32.29 | 12.57 | 8.8 | 8.68 ± 0.06 | 232.63 ± 84.49 | 2229 | 1731.4 | 5.17 |
| Isopods | final | No | 32.29 | 12.57 | 9.3 | 9.17 ± 0.05 | 176.91 ± 85.86 | 2882.6 | 1731.4 | 12.9 |
| | | | | | | | | | | |
| Isopods | final | Yes | 32.78 | 11.38 | 7.8 | 7.65 ± 0.16 | 982.49 ± 274.71 | 2050.5 | 2010.6 | 0.8 |
| Isopods | final | Yes | 32.78 | 11.38 | 8.3 | 8.18 ± 0.07 | 356.69 ± 91.8 | 2253.3 | 2010.6 | 2.6 |
| Isopods | final | Yes | 32.78 | 11.38 | 8.8 | 8.67 ± 0.05 | 234.32 ± 77.45 | 2540.4 | 2010.6 | 5.71 |
| Isopods | final | Yes | 32.78 | 11.38 | 9.3 | 9.16 ± 0.04 | 210.6 ± 142.64 | 3264.7 | 2010.6 | 14.47 |

Why was unfiltered water used in experiments? Line 126

R: Unfiltered water was used because this is what we had abundant access to at our location. Our wet lab facility has raw seawater, which comes directly from its source (the bay) on demand. Our facility also has very coarsely filtered seawater (essentially sieved). Algae and bacteria are not removed from the filtered seawater, and it sits in a holding tank until needed, so can have lower oxygen than our raw seawater. The amount of water needed each day prohibited us from manually filtering water prior to use. Text has been added to the revised manuscript at lines 152-157 and now reads: "The PNNL wet laboratory facility has abundant access to raw seawater, which comes on demand directly from the bay via the facility's seawater intake, located near the seafloor at about 10 m depth. The facility also has very coarsely filtered seawater, which sits in a holding tank until needed. This filtered seawater

however, can have lower oxygen levels than the raw seawater, and is more prone to increased temperatures inside of the tank. The amount of water needed each day prohibited us from manually filtering water prior to use. For these reasons, unfiltered water was used in the experiments."

What is the uncertainty of the YSI probe? Line 127

R: The accuracy of the probe is ±0.2 $pH_{NBS}$ units. We calibrated the probe daily before use and checked it with pH 7 buffer ensuring it was within ± 0.05 pH before use. Text has been added to the revised manuscript at lines 175-176 stating "The accuracy of the probe is ±0.2 $pH_{NBS}$ units. We calibrated the probe daily before use and checked it with pH 7 buffer ensuring it was within ± 0.05 pH before use."

How does the variability of pH (7.8 +/- 0.3) impact the amount of NaOH that needs to be added? Is it possible that water with higher initial pH, where less NaOH was added, could induce less negative effects?

R: The quantity of NaOH added is a function of the initial pH, temperature, salinity, and volume, as calculated by the CO2SYS Excel macro (Pierrot et al. 2021). So indeed, less NaOH would need to be added when starting pH was higher. These parameters were measured at each water treatment refresh in order to calculate the amount of NaOH needed.  Text has been modified and lines 178-181 now states: "Temperature, and salinity, and initial pH readings measured in the bucket, along with water volume, were used to calculate the amount of NaOH needed to reach the desired pH for each treatment group, using a CO2SYS  Excel macro (Pierrot et al., 2021). If initial pH was higher (closer to target pH), then less NaOH was needed to reach the target pH." In terms of negative effects, our hypothesis is that pH rather than alkalinity drives negative effects, and we achieved the same target pH regardless of starting pH. Within specific treatment groups, there was no correlation between the amount of NaOH added to reach the desired pH and mortality.

Was NaoH actually produced in the electrodialysis or was it commercially available?  line 133-134.

R: As stated in the manuscript L143, commercial NaOH (0.5 M, Honeywell Chemicals 352576X1L) was used in the experiment to supplement NaOH that would have been produced via electrodialysis. Text was added to the revised manuscript at lines 184-185 stating "Our facility is now capable of generating NaOH via electrodialysis, but this was not available at the beginning of the project/experiment" to provide clarification.

All OA/OAE biological studies need to report on the whole carbonate chemistry parameters and none of this is provided, not for the baseline chemistry and not for the amended (treated with NaOH) water. This absolutely needs to be added.  In the same way, all the changes are only reported in pH, while we need to understand also the change in TA to understand how much NaOH was added to different treatments. Provide a clear and methodical way of representing missing data.  Lines 177-180 are not solid justification and this needs to be amended.

R: Carbonate chemistry calculations will be added in revisions and will be included as a new table in the result section about changes in pH and $pCO_2$ (see above). We agree that it is important context for the experiments, which are primarily focused on biological responses. The table will include the amount of NaOH added and the calculated alkalinity.

Figure 1 is not clear and could be improved: 3 acclimation tanks with four treatments? How was this really conducted? Also missing is the chemical control with no NaOH added to demonstrate how the diel variability impacts diel changes in seawater before added NaOH.

R: Figure 1 has been improved for the revised version (see below). The number of acclimation tanks does not relate to the number of treatments. As explained L141-142, 5 individuals were randomly picked from each acclimation tank and moved into a jar at the start of each round of experiment. This will be clarified in the revised version and represented in the revised figure. In addition, the differences between the types of controls (non-treated water with animals, non-treated water without animals, and treated water without animals) will be clarified in the revised text at lines 197-201 by adding "Each three rounds of the experiments for both sea hares and isopods comprised a total of 24 jars: five at control pH (no NaOH added) with five animals each for a total of 25 animals, one chemical control (without animals) at control pH, five at each of the three pH treatments (NaOH added) with five animals each for a total of 25 animals per treatment, and one chemical control (without animals) for each of the three pH treatments (NaOH added).", and better represented in the revised figure. The figure caption now reads "Figure 1. Experimental setup of laboratory water table, including the 1-liter glass jars (colored circles) used in the experiment and the acclimation tanks (blue rectangles). The white circles indicate control jars with pH ≈ 7.8 (no NaOH added), yellow circles indicate low treatment with pH ≈ 8.3, orange circles indicate medium treatment with pH ≈ 8.8, and red circles indicate high treatment with pH ≈ 9.3. The dashed circles with "C" indicate the chemical control jars in which only treated, or control, seawater was added without the presence of sea hares or isopods. Animals were distributed randomly from the acclimation tanks to the experiment jars, as indicated by the black arrows. The circles "W" indicates the water input, the circle with "D" indicates the drain on the water table, and the blue arrows indicate the water flow along the table."

[Figure]

Why was pCO2 then measured if full carb chem was not calculated and provided (the variability in such big pH changes should still be theoretically lower than the uncertainty of the measurements).

R: As stated previously, we plan to include a summary of carbonate speciation calculations derived from $pCO_2$ and pH (see above). We intend to describe this data to provide context for experimental conditions that the organisms experienced.

A major drawback is reporting pooled data on water chemistry as well as on the experimental results. You should provide unpooled data and conduct analyses on this? How else was the variability determined that the within the treatment levels?

R: Unpooled data will now be provided in a table as supplementary material. However, due to the low numbers, data were pooled for the purpose of statistical analyses. Within treatment level variability was determined one-way ANOVAs.

Please, explain table 1 in more detail, I do not understand how min, max and average pH/T/DO levels determined the treatments? Why are there three values- is it form the diel variability? If so, why was the water not always taken at the same time to avoid this variability? How does this impact NaOH additions. The variability on the initial pH levels is almost as high as between treatment 1 and 2.

R: Because of the time it took to refresh the water for all 24 jars, the water drawn over 3h was subject to diel and tidal variability. Seawater was pumped from the bay directly by our facility, and the water we receive is subject to environmental variability. Data from each jar was taken into account to calculate how much NaOH was necessary to add to reach the desired treatment pH. The table caption has been clarified in the revised version to read as "Table 1. Because of the diel and tidal variability in water quality over the time the water was drawn to refresh the jars each day, this table summarizes the overall range in pH, salinity, dissolved oxygen, and temperature of ambient seawater experienced within each round of experiment before NaOH treatment was added to each treatment group."

**Results section:**

Why are the changes in pCO2 and pH are not a major part of the experiments in linking chemistry to the biological data? Strongly consider removing after all the parameters have been provided.

R: As mentioned above, we will add carbonate speciation calculations to address the reviewer's previous comments. Based on this comment, we will consider noting carbonate speciation in the methods section as experimental context rather than presenting chemical data as results. We thank the reviewer for this useful suggestion.

Why was there such an increase in pCO2? Line 201-204 inaccurate statements

R: The mason jars were airtight and contained active microbial communities. Although there is normal indoor lighting in our wet labs, we did not subject the jars to grow lights, nighttime lighting, or natural lights (it is a windowless lab). Therefore, algal production likely slowed/stopped while microbial respiration continued, resulting in $CO_2$ production and $O_2$ consumption. Likewise, the animals were actively respiring until dead, producing elevated $pCO_2$ and lowering pH. This is why we refreshed the water twice per day to maintain pH near our target treatments. This description will be added to the text in the results section.

From the Figure 2 is appears that in most rounds the treatments 8.3, 8.8 and 9.3 were the same (accounting for the variability) and there were no statistical differences between them, only 7.8 was different (not in Round 1 for no hares present, and in Round 1 and 2 for hares present). The same issue of the treatment overlap was present in the isopod treatments. This likely is the consequence of a not properly tight system. Can you provide pH data for all the rounds?

R: As explained above, this is probably a result of uncontrolled microbial activity in the jars. pH data for all water refreshes will be provided in the revised version as supplementary material.

Given the overlap in experimental treatments, it is impossible to separate the effects on the treatment levels. First, the levels need to be examined to really understand which of them are sufficiently different and only then examine biological differences. Right now section 3.2 is not valid. As it seems, the combined 8.3, 8.8 and 9.3 could be one treatment, which could be compared to 7.8, which means a 2-treatment design (at the best).

R: The treatments we were interested in testing were the levels of pH, which did not overlap even with the YSI error of +/- 0.2 in which treatments could fall into the ranges of 8.1-8.5, 8.6-9, and 9.1-9.5. In the revised manuscript we will provide a table in the supplementary material of all measured pH values to show that these were separate treatments.

What is mortality (LC50) or LOEC? This is an ecotox study so present all data, i.e. NOEC; LOEC, EC50 on the combined graphs (not only LC50).

R: Text has been added to the revised manuscript at lines 456-458 stating "The lowest observed effect concentration (LOEC) for sea hares was pH of 8.8, and the no observed effect concentration (NOEC) for sea hares was pH of 8.3. For isopods, LOEC and NOEC could not be determined because no mortality in any treatment group was statistically higher than mortality in the control group."

I am not sure why all the regression models (Table 3) were built with multiple parameters 8salinity, DO, temp) if only the pH (CO2) was intentionally (artificially) changed?

R: Our intention was to examine if drivers other than our intended manipulation (pH) caused mortality. For example, did a lack of oxygen, difference in temperature, or change in salinity confound our results? The lack of correlation with parameters other than pH confirm that pH indeed was the cause of mortality differences across treatments assuming malnourishment equally affected each pH treatment. This will be clarified in our revised manuscript: "Multiple linear regression models were used in conjunction with the boxplots to determine whether pH was the driving factor behind mortality, or whether changes in salinity, DO levels, or temperature may have confounded the results."

Until these changes are introduced, the text from line 380 is not applicable (I did not continue from here onwards).

R: Instead of LC50, the figures and captions will be revised to present and explain the exposure time to each treatment after which 50% mortality occurs. We will also present NOEC (no observed effect concentration) and LOEC (lowest observed effect concentration), using pH as a concentration and clarifying this in the revised text. Text has been added to lines 456-458 that states: "The lowest observed effect concentration (LOEC) for sea hares was pH of 8.8, and the no observed effect concentration

(NOEC) for sea hares was pH of 8.3. For isopods, LOEC and NOEC could not be determined because no mortality in any treatment group was statistically higher than mortality in the control group."

Why is data discussed in the Discussion (growth and reproductive behaviors) not presented in the Results?

R: These observations will be added to the results in the revised version.

Discussion should be expanded!

R: Once introduction, methods, and results are revised (see various proposed revisions above), the discussion will be expanded to reflect the revised information: e.g., LOEC and NOEC in addition to LC50, applicability of ecotox experiments to OAE effects, direct application to OAE field trials (continuous exposure in mixing zone).